

# Cryptic diversity among Yazoo Darters (Percidae: *Etheostoma raneyi*) in disjunct watersheds of northern Mississippi

Ken A. Sterling[1], Stuart V. Nielsen[2], Andrew J. Brown[3], Melvin L. Warren, Jr.[1] and Brice P. Noonan[4]

[1] USDA Forest Service, Southern Research Station, Stream Ecology Laboratory, Oxford, MS, United States of America
[2] Division of Herpetology, Florida Museum of Natural History, Gainesville, FL, United States of America
[3] Louisiana Purchase Gardens and Zoo, Monroe, LA, United States of America
[4] Department of Biology, University of Mississippi, University, MS, United States of America

Corresponding author
Ken A. Sterling,
kenneth.a.sterling@usda.gov

## ABSTRACT

The Yazoo Darter, *Etheostoma raneyi* (Percidae), is an imperiled freshwater fish species endemic to tributaries of the Yocona and Little Tallahatchie rivers of the upper Yazoo River basin, in northern Mississippi, USA. The two populations are allopatric, isolated by unsuitable lowland habitat between the two river drainages. Relevant literature suggests that populations in the Yocona River represent an undescribed species, but a lack of data prevents a thorough evaluation of possible diversity throughout the range of the species. Our goals were to estimate phylogenetic relationships of the Yazoo Darter across its distribution and identify cryptic diversity for conservation management purposes. Maximum likelihood (ML) phylogenetic analyses of the mitochondrial cytochrome *b* (*cytb*) gene returned two reciprocally monophyletic clades representing the two river drainages with high support. Bayesian analysis of *cytb* was consistent with the ML analysis but with low support for the Yocona River clade. Analyses of the nuclear *S7* gene yielded unresolved relationships among individuals in the Little Tallahatchie River drainage with mostly low support, but returned a monophyletic clade for individuals from the Yocona River drainage with high support. No haplotypes were shared between the drainages for either gene. Additional cryptic diversity within the two drainages was not indicated. Estimated divergence between Yazoo Darters in the two drainages occurred during the Pleistocene (<1 million years ago) and was likely linked to repeated spatial shifts in suitable habitat and changes in watershed configurations during glacial cycles. Individuals from the Yocona River drainage had lower genetic diversity consistent with the literature. Our results indicate that Yazoo Darters in the Yocona River drainage are genetically distinct and that there is support for recognizing Yazoo Darter populations in the Yocona River drainage as a new species under the unified species concept.

## INTRODUCTION

The southeastern United States has a globally significant amount of diversity among its freshwater fishes (*Abell et al., 2008*). A large portion of this diversity is contained within

Etheostomatinae (Percidae), the darters (*Jelks et al., 2008*; *Page & Burr, 2011*). Though the group shows a wide variety of life history strategies and associated distributional patterns (*Fluker, Kuhajda & Harris, 2014*), many species of darters are range-limited (microendemics) (*Page, 1983*; *Page & Burr, 2011*) and share a suite of life history characteristics that are associated with limited dispersal (*Turner & Trexler, 1998*; *Turner, 2001*), including niche conservatism (*Keck & Near, 2010*). The discovery of microendemism in darters is occurring more frequently because, at least in part, the routine use of genetic tools is increasingly uncovering cryptic diversity (*Hollingsworth Jr & Near, 2009*; *April et al., 2011*; *Echelle et al., 2015*; *Kozal et al., 2017*; *Matthews & Turner, 2019*).

The Yazoo Darter (*Etheostoma raneyi* Suttkus and Bart, 1994) is a snubnose darter (clade Adonia, *sensu Near et al., 2011*) distributed in the upper Yazoo River basin in north-central Mississippi (Figs. 1 and 2, Figs. S1–S3). Surface geology mostly comprises highly erodible, unconsolidated sands and clays with resulting fine substrates within streams. Topography is relatively flat compared with upland regions but is more variable compared with the Lower Gulf Coastal Plain and Mississippi Alluvial Plain to the west (*Ross, 2001*; *Keck & Etnier, 2005*; *Powers & Warren Jr, 2009*) (Fig. 1). Yazoo Darters occur in headwater tributaries of the Little Tallahatchie (L.T.R.) and Yocona (Y.R.) rivers whose confluence lies in bottomland habitat of the Mississippi Alluvial Plain, which is unfavorable for the darter. In common with other snubnose darters, Yazoo Darters are small (<65 mm Standard Length), benthic insectivores lacking a swim bladder (*Page, 1983*; *Johnston & Haag, 1996*; *Sterling, Warren Jr & Henderson, 2013*). Long distance movements for spawning or feeding are not documented for snubnose darter species. Larvae of snubnose darters, including the Yazoo Darter, are pelagic but active swimmers upon hatching and select for sheltered areas out of direct current immediately downstream of spawning areas; passive drift of larvae is not documented (*Simon & Wallus, 2006*; *Ruble, Sterling & Warren Jr, 2019*). A population genetic study of the Yazoo Darter using microsatellite data indicated limited historical dispersal among tributary streams and virtually no contemporary dispersal, likely because of anthropogenic habitat destruction (*Warren Jr, Haag & Adams, 2002*; *Sterling et al., 2012*). Genetic structure was high across small spatial scales among some tributary populations ($F_{st} = 0.03$–$0.17$) within each major drainage where the species occurs (L.T.R. and Y.R.) and was also high between drainages ($F_{st} = 0.17$–$0.29$) (*Sterling et al., 2012*).

A phylogenetic analysis of Upper Gulf Coastal Plain snubnose darters (*Etheostoma pyrrhogaster*, *E. cervus*, and *E. raneyi*, see Fig. 1) in western Kentucky, Tennessee, and northern Mississippi indicated that Yazoo Darters inhabiting the L.T.R. and Y.R were genetically distinct and reciprocally monophyletic with high posterior support. *Powers & Warren Jr, (2009)* suggested that the same vicariant events isolated all forms of darters they examined in the Upper Gulf Coastal Plain. However, the study was limited to six Yazoo Darters from only a few streams in each drainage ($n = 12$) (*Powers & Warren Jr, 2009*).

The Yazoo Darter is categorized as vulnerable by the American Fisheries Society (*Jelks et al., 2008*) and the Southeastern Fishes Council (*Warren Jr et al., 2000*), as globally imperiled by the Nature Conservancy (*NatureServe, 2019*), as sensitive by the USDA Forest Service (*USDA Forest Service, 2013*), and as a Tier 1 species of greatest conservation need by the Mississippi State Wildlife Action Plan (*Mississippi Museum of Natural Science, 2015*).

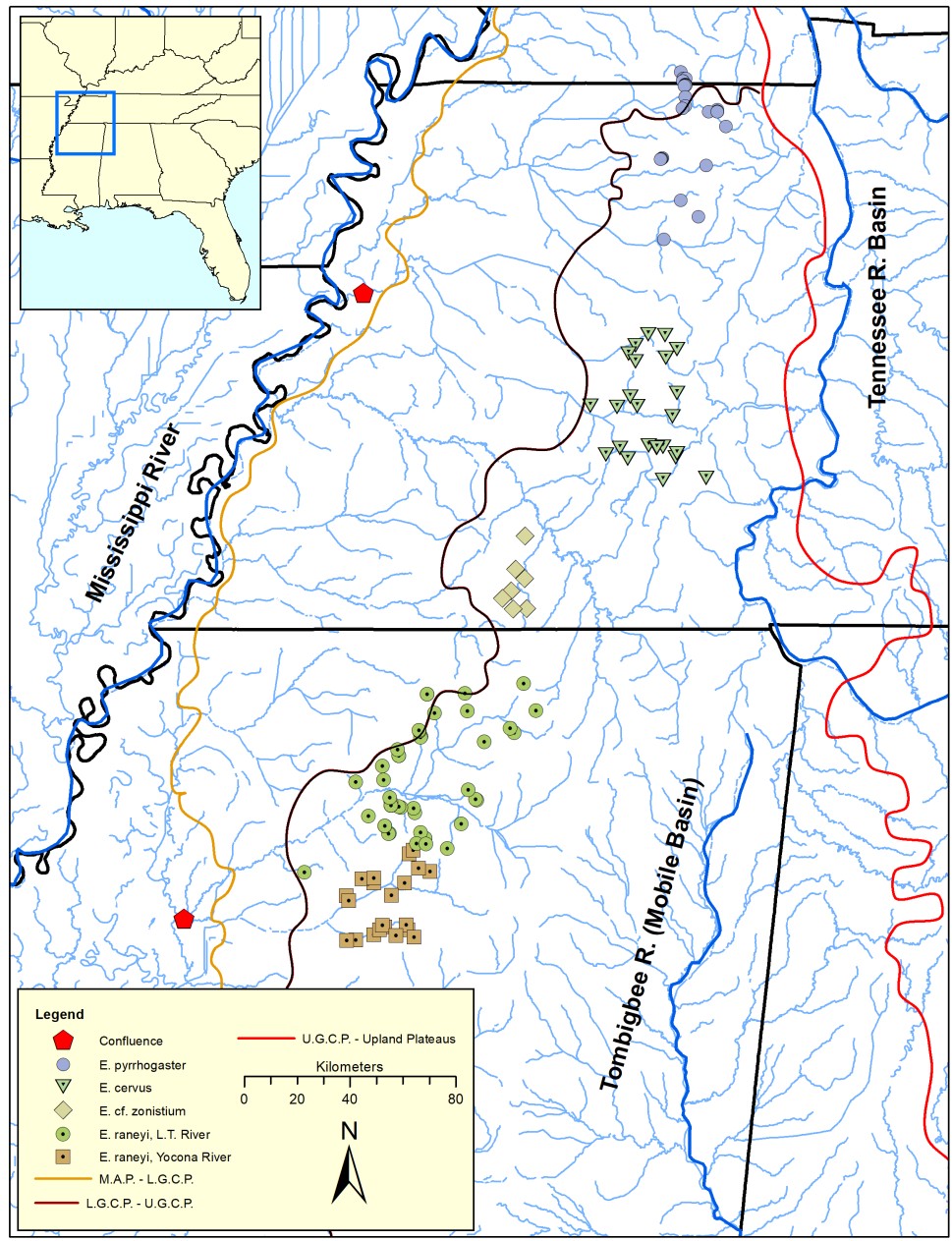

**Figure 1** **Distribution of snubnose darters among lower Mississippi River drainages of Kentucky, Tennessee, and Mississippi (southeastern United States).** Major river systems and physiographic provinces discussed in the text are shown; abbreviations are defined as: L.T. = Little Tallahatchie, M.A.P. = Mississippi Alluvial Plain, L.G.C.P. = Lower Gulf Coastal Plain, U.G.C.P. = Upper Gulf Coastal Plain.

Human-assisted gene flow among tributaries within each drainage was recommended as a conservation management action (*Sterling et al., 2012*). Even so, an investigation of possible cryptic diversity across the species' distribution within each drainage as well as estimates of genetic structure using markers reflecting deeper evolutionary relationships is needed to better inform such an action. We used genetic sequences from mitochondrial *cytb* and

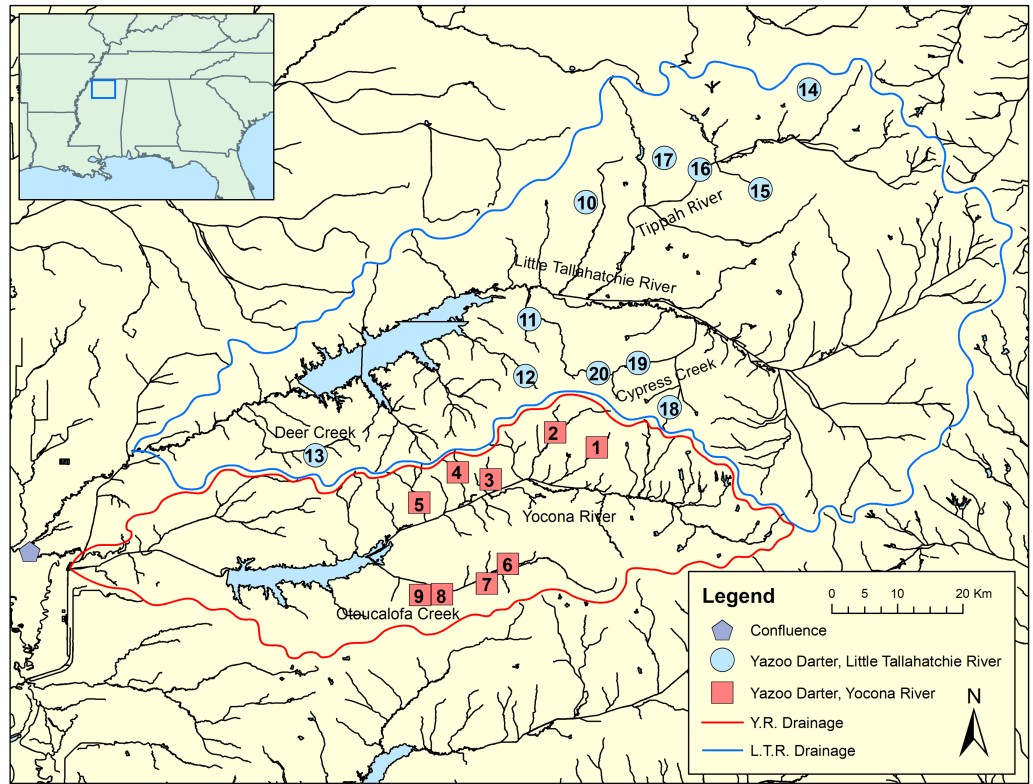

**Figure 2 Map showing genetic tissue sample sites for Yazoo Darters in the Little Tallahatchie River drainage (blue) and Yocona River drainage (red).** Names of watersheds used for genetic distance estimates (see Table 3) and discussed in the text are also shown. Numbers correspond to data in Table 1; Y.R. = Yocona River, L.T.R. = Little Tallahatchie River.

nuclear *S7* genes, to (1) investigate possible cryptic diversity within and between each major drainage; (2) estimate phylogenetic relationships among populations within each drainage to inform discussion of human-assisted gene flow for conservation management; and (3) to assess the results from *Powers & Warren Jr (2009)* using larger sample sizes from sites across the distribution of the species.

## MATERIAL AND METHODS

We sampled 117 individuals from 20 streams representative of the entire range of the Yazoo Darter via single-pass backpack electrofishing, dip nets, and seines. Collecting localities included nine streams in the Y.R. drainage and 11 streams in the L.T.R. drainage (Fig. 2; Table 1, Tables S1 and S2). We obtained tissue samples by either taking pelvic fin clips or by collecting voucher specimens, which we stored in 95% ethanol at −74 °C. This study was conducted with the approval of the University of Mississippi IACUC Committee (protocol 09-027), using annual collection permits issued to us from the Mississippi Museum of Natural Science (2009–2017: 0604091, 0513101, 0624112, 0622122, 0602132, 0610142, 0624151, 0715163, 1010173).

**Table 1 Genetic tissue sample data for each of the two major drainages within the distribution of the Yazoo Darter.** Sample locations, drainage, and sample sizes for genetic analyses are shown. Site ID numbers correspond to Fig. 2; see Tables S1 and S2; U.T., unnamed tributary.

| Site ID | Drainage | Stream | *Cytb*, n | *S7*, n | Latitude | Longitude |
|---|---|---|---|---|---|---|
| 1 | Yocona River | Pumpkin Creek | 4 | 1 | 34.327 | −89.398 |
| 2 | Yocona River | Yellow Leaf Creek | 2 | 1 | 34.348 | −89.455 |
| 3 | Yocona River | Morris Creek | 4 | 4 | 34.283 | −89.544 |
| 4 | Yocona River | Taylor Creek | 5 | 5 | 34.293 | −89.589 |
| 5 | Yocona River | Splinter Creek | 3 | 4 | 34.251 | −89.642 |
| 6 | Yocona River | Mill Creek | 6 | 6 | 34.167 | −89.52 |
| 7 | Yocona River | Gordon Branch | 3 | 2 | 34.14 | −89.549 |
| 8 | Yocona River | U.T. Otoucalofa Creek | 4 | 3 | 34.125 | −89.611 |
| 9 | Yocona River | Johnston Creek | 6 | 6 | 34.124 | −89.641 |
| 10 | Little Tallahatchie River | Big Spring Creek | 10 | 13 | 34.664 | −89.413 |
| 11 | Little Tallahatchie River | Graham Mill Creek | 3 | 3 | 34.503 | −89.491 |
| 12 | Little Tallahatchie River | Hurricane Creek | 2 | 3 | 34.425 | −89.496 |
| 13 | Little Tallahatchie River | Deer Creek | 6 | 6 | 34.316 | −89.785 |
| 14 | Little Tallahatchie River | Yellow Rabbit Creek | 5 | 4 | 34.819 | −89.106 |
| 15 | Little Tallahatchie River | Chilli Creek | 5 | 4 | 34.682 | −89.173 |
| 16 | Little Tallahatchie River | U.T. Tippah River | 2 | 2 | 34.709 | −89.256 |
| 17 | Little Tallahatchie River | Chewalla Creek | 4 | 3 | 34.725 | −89.305 |
| 18 | Little Tallahatchie River | Cypress Creek | 7 | 6 | 34.382 | −89.298 |
| 19 | Little Tallahatchie River | Puskus Creek | 12 | 6 | 34.443 | −89.341 |
| 20 | Little Tallahatchie River | Bay Springs Branch | 2 | 1 | 34.429 | −89.396 |

We isolated whole genomic DNA (*MacManes, 2013*) and used previously published PCR primers to amplify the entire mitochondrial *cytb* gene (1,140 bp; *Song, Near & Page, 1998*) and the forward sequence of intron 1 of the nuclear *S7* ribosomal gene (599 bp; *Chow & Hazama, 1998*). PCR components were as follows: 9.8 µl ddH$_2$O, 0.2 µl dNTP, 0.4 µl MgCl$_2$, 2 µl 5x reaction buffer, 0.2 µl each 10 nM primer, 0.15 µl Phire$^{TM}$ *Taq*, and 1.5 µl of template DNA (~15 µl total reaction volume). We set conditions for PCR reactions as 98 °C for 30 s, followed by 30 cycles of 98 °C for 6 s, 53.1−56 °C for 30 s, and 72 °C for 60 s. We purified and sequenced PCR products using ExoSAP-IT (ThermoFisher Scientific) and Big Dye (ver. 3.1, ThermoFisher Scientific) according to manufacturer's recommendations. Arizona State University DNA sequencing facility processed the samples (https://asu.corefacilities.org/service_center/show_external/3900/asu-dna-lab) using an automated ABI 3730 sequencer. We assembled all resulting forward and reverse sequences into contigs for each individual and aligned them using MEGA (ver. 7.0.26; *Kumar, Stecher & Tamura, 2016*). We obtained outgroup sequence data and sequences for two additional Yazoo Darters (*Near et al., 2011*), one from each major drainage, from GenBank for use in our analyses. Sequence data for this study are available from GenBank (https://www.ncbi.nlm.nih.gov/nuccore/) and Dryad (*Kozal et al., 2017*) (Tables S1–S3).

Data from *cytb* and *S7* could not be combined into a single concatenated analysis because the data were not derived from the same set of individuals (*Matthews & Turner, 2019*). We used PartitionFinder V 1.1.1 (*Lanfear et al., 2012*) to find the best-fit model for

each locus. The *cytb* dataset was partitioned by 1st, 2nd, and 3rd codon positions, and the *S7* dataset was analyzed as a single partition. We analyzed partitioned datasets for each gene (Tables S1, and S2) using Bayesian Inference (BI) implemented in MrBayes ver. 3.2.6 (*Ronquist et al., 2012*) via CIPRES Science Gateway ver. 3.3 (https://www.phylo.org/) (*Miller, Pfeiffer & Schwartz, 2010*). Each partition/analysis included the most appropriate substitution models for the two loci as suggested by PartitionFinder. We used two runs of MrBayes for $10^6$ generations; four Markov chains sampled every 10,000 steps and Tracer (ver. 1.7.1; *Rambaut et al., 2018*) removed 25% of the posterior trees as burn-in. We then generated a 50% majority rule consensus tree in MrBayes. We used the same data to construct Maximum Likelihood (ML) trees using RAxML-HPC ver. 8.0 (https://cme.h-its.org/exelixis/web/software/raxml/) (*Stamatakis, 2014*) also using the CIPRES Science Gateway ver. 3.3 (*Miller, Pfeiffer & Schwartz, 2010*). We used the default GTR model and performed 100 bootstrap replicates to assess nodal support. We considered nodes with posterior probabilities ≥95% as strongly supported (*Huelsenbeck & Ronquist, 2001*).

We visualized relationships among individuals using haplotype networks (TCS v. 1.21; *Clement, Posada & Crandall, 2000*) for each gene. We estimated uncorrected pairwise genetic distances (p-distances) using MEGA ver. 7.0.26 (*Kumar, Stecher & Tamura, 2016*) between drainages and among watersheds within drainages for each gene. For comparison, we also generated p-distances among all snubnose darters (clade Adonia, *sensu Near et al., 2011*) using our data and publicly available *cytb* genetic sequences (see Table S4 for genetic sequence data).

We calculated the number of haplotypes, and haplotype diversity for both loci using DNAsp V 5.10 (*Librado & Rozas, 2009*) between drainages and among watersheds within drainages. We calculated estimates of divergence times using rates of molecular evolution for the *cytb* (1.8%/my) and *S7* (0.34%/my) genes reported by *Near et al. (2011)*, and our observed genetic distance values produced by MEGA.

## RESULTS

The most appropriate substitution models for the 1st, 2nd, and 3rd codon positions of the *cytb* (1139 nucleotides (nt) in length) were F81, GTR+G, and K80+I and for the *S7* gene (530 nt), F81+G. Results from Bayesian and ML analyses for *cytb* indicate two monophyletic clades congruent with the two river drainages (Figs. 3 and 4, Figs. S4 and S5). Support for reciprocally monophyletic clades was high for the ML analysis (bootstrap support: Y.R., 95%; L.T.R., 100%), but only weakly supported for the Bayesian analysis (posterior probabilities: Y.R., 0.12; L.T.R., 1.0). Results from the *S7* data (Figs. 5 and 6, Figs. S6 and S7) indicated weakly supported and inconsistent phylogenetic relationships among individuals from the L.T.R. drainage, though samples from the Y.R. drainage composed a single clade with high support (95% bootstrap support and 0.97 posterior probability). Haplotype networks for *S7* and *cytb* indicate that no haplotypes were shared between drainages (Fig. 7). A total of fifteen genetic characters from both genes are diagnostic of Yazoo Darters in the two major river drainages (Table 2).

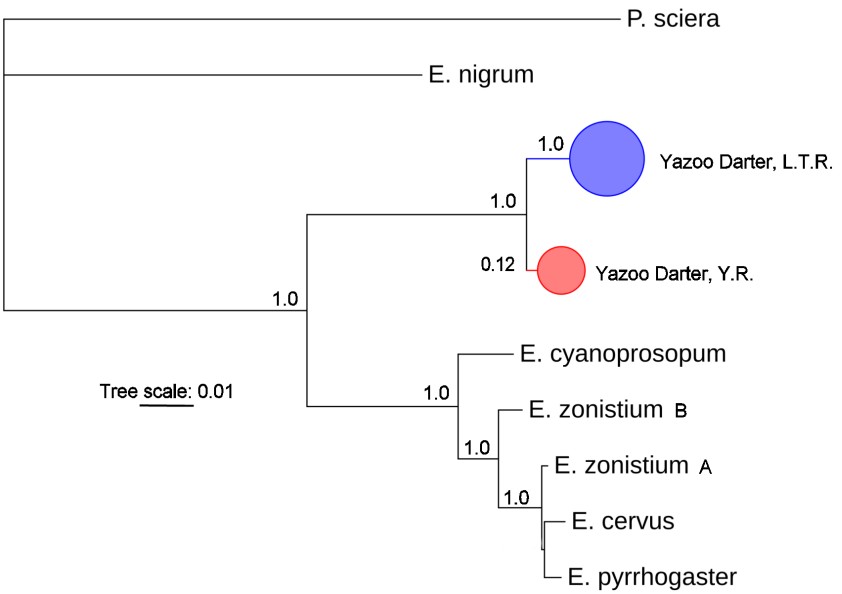

**Figure 3 Phylogenetic tree of the partitioned *cytb* dataset using Bayesian estimation (MrBayes ver. 3.2.6).** Bayesian posterior probabilities ≥0.95 are shown (except for the Yocona River clade) at the nodes (see Table S1 for sequence data); bubble sizes for the pruned nodes are proportional to sample size; L.T.R. = Little Tallahatchie River drainage, Y.R. = Yocona River drainage.

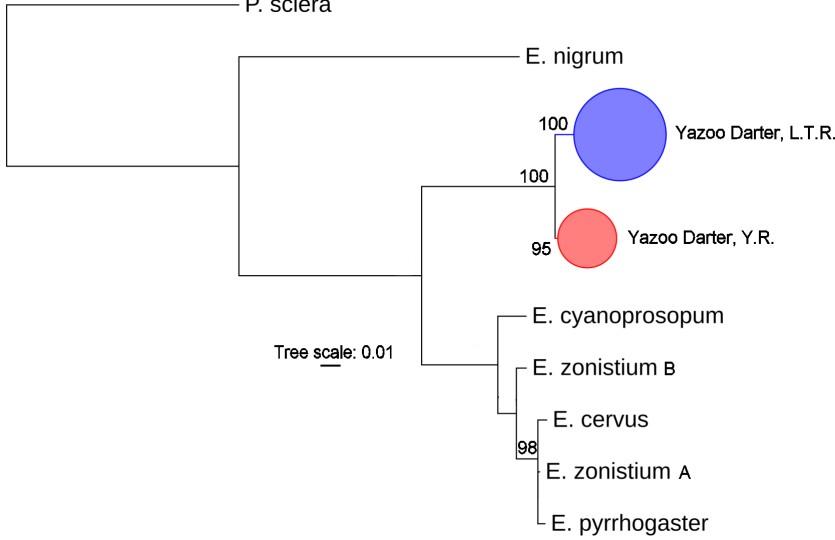

**Figure 4 Phylogenetic tree of the partitioned *cytb* dataset using maximum likelihood estimation (RAxML-HPC ver. 8.0).** Bootstrap values ≥95 are shown at the nodes (see Table S1 for sequence data); bubble sizes for the pruned nodes are proportional to sample size; L.T.R. = Little Tallahatchie River drainage, Y.R. = Yocona River drainage.

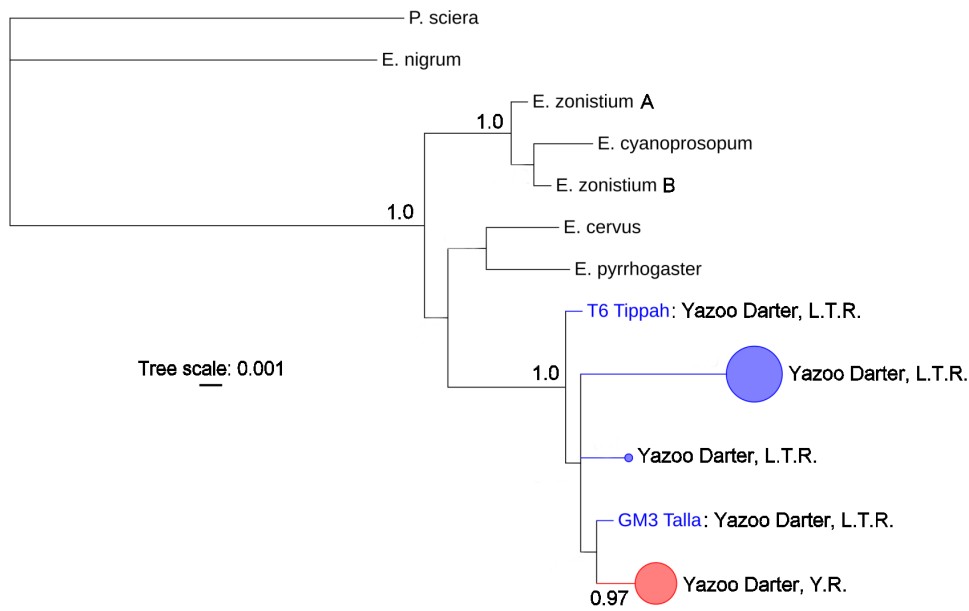

**Figure 5** **Phylogenetic tree of the partitioned *S7* dataset using Bayesian estimation** (**MrBayes ver. 3.2.6**)**.** Bayesian posterior probabilities ≥0.95 are shown at the nodes (see Table S2 for sequence data); bubble sizes for the pruned nodes are proportional to sample size; L.T.R. = Little Tallahatchie River drainage, Y.R. = Yocona River drainage.

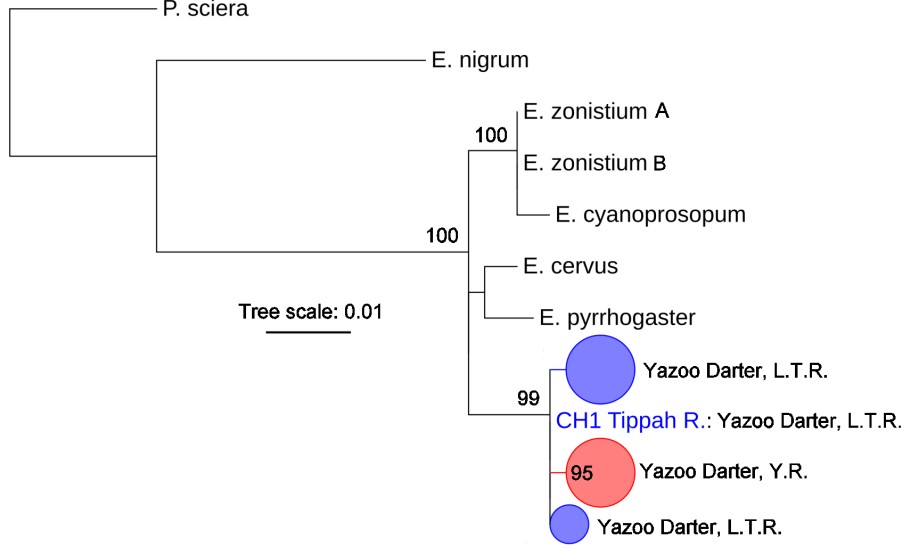

**Figure 6** **Phylogenetic tree of the partitioned *S7* dataset using maximum likelihood estimation** (**RAxML-HPC ver. 8.0**)**.** Bootstrap values ≥95 are shown at the nodes (see Table S2 for sequence data); bubble sizes for the pruned nodes are proportional to sample size; L.T.R. = Little Tallahatchie River drainage, Y.R. = Yocona River drainage.

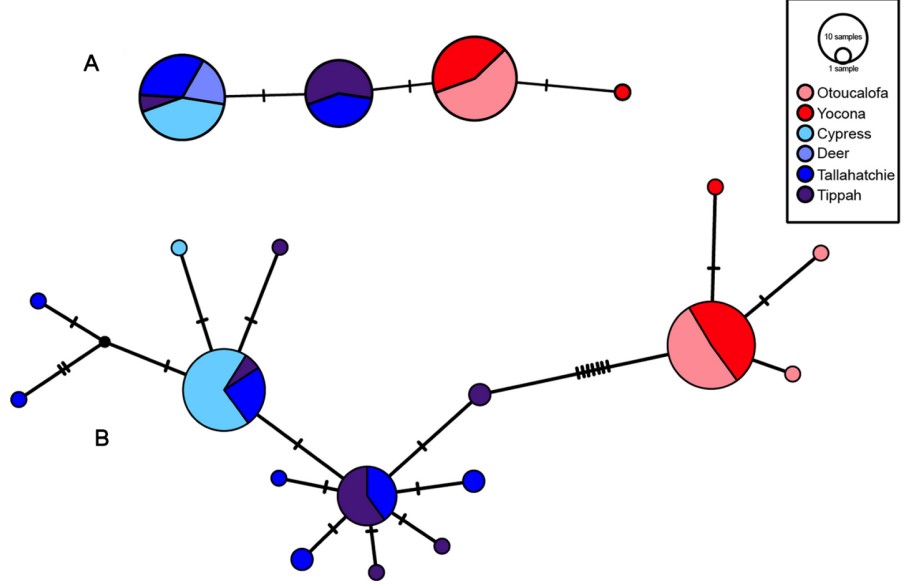

**Figure 7** S7 (A) and *cytb* (B) haplotype networks for samples among the Little Tallahatchie and Yocona River drainages, and watersheds (e.g.,Cypress Creek) within drainages (see Fig. 2). Red and blue indicates the Yocona and Little Tallahatchie river drainages, respectively.

Uncorrected p-distances for Yazoo Darters (*cytb*) between drainages were 0.8% and among watersheds within drainages was 0.01% in the Y.R. drainage and 0.1 and 0.11% in the L.T.R. drainage (Table 3). For comparison, p-distances (*cytb*) among other snubnose darters ranged from 0.5–14.53% (Tables S4–S5). P-distances for Yazoo Darters (*S7*) between drainages were 0.3% and among watersheds within drainages were 0.01% in the Y.R. drainage and ranged from 0.07–0.17% in the L.T.R. drainage (Table 3).

Haplotype diversity was higher in the L.T.R. drainage (*cytb*: Hd = 0.66, 11 haplotypes; *S7*: Hd = 0.48, 2 haplotypes) than in the Y.R. drainage (*cytb*: Hd = 0.11, 3 haplotypes; *S7*: Hd = 0.06, 2 haplotypes). Estimated times of divergence were 0.88 (*S7*) and 0.44 my (*cytb*).

## DISCUSSION

Our results indicate, (1) there is no evidence of cryptic diversity within each major river drainage; (2) genetic diversity is lower in the Y.R. drainage relative to the L.T.R. drainage; (3) consistent with the results from *Powers & Warren Jr (2009)*, there is support for recognizing Yazoo Darter populations in the Y.R. drainage as a distinct species under the unified species concept (*de Queiroz 2007*); (4) Our estimates of time of divergence are similar to estimates for closely related snubnose darter species in Tennessee and Kentucky (*Kozal et al., 2017*), which supports the proposal in *Powers & Warren Jr (2009)* that the same vicariant events led to a late Pleistocene species radiation among snubnose darters in western Tennessee and Kentucky and northern Mississippi.

**Table 2 Genetic characters that diagnose allopatric populations of Yazoo Darters in the Little Tallahatchie (L.T.R.) and Yocona rivers (Y.R.) using mitochondrial cytochrome b (*cytb*) and nuclear *S7* genes.** Numbers indicate the location of the character along the genetic sequence; A, adenine; C, cytosine; T, thymine; G, guanine.

| Character (*cytb*) | L.T.R. | Y.R. |
|---|---|---|
| 147 | A | G |
| 165 | C | T |
| 585 | C | T |
| 588 | T | C |
| 654 | A | G |
| 876 | G | A |
| 897 | A | G |
| 930 | G | A |
| 1,056 | A | G |
| 1,090 | G | A |
| 1,107 | G | A |
| 1,113 | G | A |
| **Character (*S7*)** | **L.T.R.** | **Y.R.** |
| 286 | G | A |
| insertion: 478 | G | – |
| insertion: 479 | C | – |

**Table 3 Uncorrected pairwise genetic distances (p-distance, %) among watersheds for Yazoo Darters.** Values for *cytb* are below the diagonal and for *S7* are above the diagonal; abbreviations are defined in the table.

| | Otoucalofa Cr. | Y.R. | L.T.R. | Tippah R. | Cypress Cr. |
|---|---|---|---|---|---|
| Otoucalofa Creek | | 0.01 | 0.35 | 0.24 | 0.41 |
| Yocona River | 0.01 | | 0.36 | 0.25 | 0.43 |
| Little Tallahatchie River | 0.82 | 0.81 | | 0.13 | 0.07 |
| Tippah River | 0.76 | 0.75 | 0.11 | | 0.17 |
| Cypress Creek | 0.83 | 0.82 | 0.10 | 0.10 | |

Though the lack of genetically distinct clades within drainages was not a surprise, the lack of a clear and consistent link between genetic clades and geography within drainages was unexpected (Fig. 2, Figs. S4–S7). Though samples from watersheds within drainages show a weak to moderate tendency to be grouped within clades (e.g., Cypress Creek, Tippah River, Otoucalofa Creek), the low support at most within-drainage nodes (phylogenetic trees) indicates that the only inference that can be made with any confidence is that our data did not reveal genetically distinct populations or cryptic diversity within either drainage. In contrast, microsatellite markers indicated that tributaries do contain genetically distinct populations with a strong isolation by distance effect (*Sterling et al., 2012*). This is explained by higher mutation rates among microsatellite markers and, to some extent, the effects of human habitat alteration and consequent isolation. Taken together, patterns of gene flow within drainages apparently have not been stable over enough generations to show a clear relationship between watersheds and genetic subclades in the *cytb* and *S7* data.

Our results show that genetic divergence and diversity is much lower in the Y.R. drainage than in the L.T.R. drainage (Table 3; Fig. 7). Genetic distances between Otoucalofa Creek and the Y.R. are an order of magnitude lower than the distances between Cypress Creek and the Tippah River, for example (Fig. 2), and the haplotype network results (Fig. 7) clearly show less genetic diversity in the Y.R. drainage. Lower divergence can be explained by the difference in area of distribution of the Yazoo Darter between the two drainages (Fig. 2). Smaller area of distribution in the Y.R. drainage (smaller watersheds with fewer and smaller streams) would likely result in greater gene flow, lower divergence, and less genetic diversity (Frankham, 1996). Another possible explanation for lower diversity in the Y.R. drainage is a founder effect, though our results are not consistent with this explanation (e.g., stochastic processes likely would have resulted in greater divergence between drainages than we observed) and do not indicate any mechanism for such an explanation (e.g., stream capture). The lower genetic diversity observed in the Y.R. drainage is consistent with previous genetic studies (Powers & Warren Jr, 2009; Sterling et al., 2012). Our results and the low effective population sizes reported in Sterling et al. (2012) indicate that human-assisted gene flow is warranted.

Genetic distances between drainages were low but are comparable to other closely related snubnose darters (Tables 3 and 4, Table S5). This is especially true for the Bandfin Darter group (Etheostoma zonistium, E. cervus, E. pyrrhogaster, and E. cf. zonistium). Similar genetic distances are almost certainly linked to the similar estimates for times of divergence among these taxa, which are recent (Kozal et al., 2017). Our observed distances are also similar to those reported for other sister species pairs of fishes (Johns & Avise, 1998).

The lack of resolution and consistency in phylogenetic clades showing relationships in the L.T.R. drainage using the S7 marker (Figs. 5 and 6) is not surprising because in young clades of darters cytb usually has more power to resolve relationships with higher support relative to nuclear genetic markers with slower mutation rates and higher effective population sizes (Avise, 2004; Keck & Near, 2008). The lack of resolution for the S7 results is consistent with other studies (Keck & Near, 2010; Echelle et al., 2015; Kozal et al., 2017). Even so, the S7 analyses did support a monophyletic clade for samples from the Y.R. drainage, and this might be explained by smaller populations in a smaller watershed as outlined earlier when discussing relative levels of genetic divergence and diversity.

Our results using cytb did produce consistent phylogenetic trees with monophyletic clades for each drainage. However, support for the monophyly of the Y.R. clade using Bayesian analysis was weak, which is odd considering the high support for the ML and S7 analyses. The low support for this clade has no apparent biological explanation. Even so, the S7 and cytb results indicate recent divergence between Yazoo Darter populations in the L.T.R. and Y.R. drainages. Divergence is supported by the lack of evidence for gene flow between the drainages, even for the samples from Deer Creek which are closest to the confluence of the L.T.R. and Y.R. drainages (Fig. 2). No haplotypes were shared between drainages, and all individuals sorted into clades consistent with the drainages from which they were sampled for all phylogenetic trees (Figs. S4–S7). This is consistent with the literature (Powers & Warren Jr, 2009; Sterling et al., 2012).

**Table 4 Uncorrected pairwise genetic distances (*p*-distances, %), among snubnose darters that are most closely related to the Yazoo Darter** (*Near et al., 2011*). Species complexes are grouped following *Near et al. (2011)*; labels for undescribed species follows *Jelks et al. (2008)*. Bold type and asterisk = values <2.0%; L.T.R., Yazoo Darter, Little Tallahatchie River drainage, Y.R., Yazoo Darter, Yocona River drainage, Fk., Fork; see Table S3 for complete results.

| | E. raneyi (Yazoo Darter group) | | E. zonistium (Bandfin Darter group) | | | | |
|---|---|---|---|---|---|---|---|
| | Y.R. | L.T.R. | E. zonistium | E. cf. zonistium | E. cervus | E. pyrrhogaster | E. cyanoprosopum |
| Y.R. | | | | | | | |
| L.T.R. | **0.75\*** | | | | | | |
| E. zonistium | 7.61 | 8.07 | | | | | |
| E. cf. zonistium | 8.33 | 8.64 | **1.29\*** | | | | |
| E. cervus | 8.45 | 8.97 | **1.42\*** | **0.50\*** | | | |
| E. pyrrhogaster | 8.61 | 9.04 | **1.44\*** | **0.72\*** | **0.86\*** | | |
| E. cyanoprosopum | 8.84 | 9.24 | 4.25 | 4.75 | 4.86 | 4.84 | |
| E. bellator | 8.50 | 8.99 | 8.93 | 9.38 | 9.48 | 9.39 | 9.84 |
| E. chermocki | 8.22 | 8.71 | 8.74 | 9.19 | 9.29 | 9.20 | 9.62 |
| "Locust Fork" | 9.70 | 10.18 | 9.64 | 9.81 | 9.80 | 9.81 | 10.24 |
| "Sipsey" | 10.29 | 10.40 | 10.69 | 11.02 | 10.99 | 10.84 | 11.02 |
| "Conasauga" | 8.79 | 8.71 | 9.03 | 9.32 | 9.41 | 9.46 | 9.51 |
| "Amicalola" | 7.84 | 7.95 | 7.78 | 8.08 | 8.17 | 8.41 | 8.20 |
| E. brevirostrum | 8.79 | 8.71 | 9.22 | 9.51 | 9.60 | 9.68 | 9.78 |
| E. simoterum | 14.33 | 14.24 | 15.03 | 15.01 | 15.04 | 15.42 | 15.01 |
| Percina sciera | 16.91 | 16.81 | 16.81 | 17.30 | 17.46 | 17.47 | 17.11 |

| | E. bellator (Warrior Darter group) | | | | E. brevirostrum (Holiday Darter group) | | |
|---|---|---|---|---|---|---|---|
| | E. bellator | E. chermocki | "Locust Fk." | "Sipsey" | "Conasauga" | "Amicalola" | E. brevirostrum |
| E. bellator | | | | | | | |
| E. chermocki | **0.57\*** | | | | | | |
| "Locust Fork" | 5.11 | 4.92 | | | | | |
| "Sipsey" | 6.24 | 6.05 | 6.57 | | | | |
| "Conasauga" | 8.79 | 8.60 | 9.22 | 9.84 | | | |
| "Amicalola" | 7.45 | 7.07 | 7.69 | 8.21 | 3.15 | | |
| E. brevirostrum | 8.88 | 8.69 | 9.89 | 10.12 | **1.05\*** | 3.63 | |
| E. simoterum | 14.33 | 14.14 | 13.90 | 14.52 | 14.90 | 13.94 | 14.71 |
| Percina sciera | 15.85 | 15.57 | 15.62 | 17.03 | 17.77 | 17.00 | 17.86 |

The vicariant events and mechanisms that led to isolation of ancestral populations and ensuing divergence among Upper Gulf Coastal Plain snubnose darters currently recognized as *E. cervus*, *E. pyrrhogaster*, and *E. zonistium* almost certainly were a factor in divergence of ancestral Yazoo Darters in the upper Yazoo River basin (*Powers & Warren Jr, 2009*; *Kozal et al., 2017*) (Fig. 1). This is based on similar estimated times of divergence among these closely related fishes, as well as similar surface geology, topography, and watershed configurations among them (*Warren Jr, Haag & Adams, 2002*; *Keck & Etnier, 2005*; *Powers & Warren Jr, 2009*; *Smith et al., 2009*; *Near et al., 2011*; *Kozal et al., 2017*). Though we restrict our discussion to the Yazoo Darter, we believe that our interpretations are generally applicable to these other species. We propose that spatial shifts in suitable

habitat for Yazoo Darters during repeated glacial cycles over the last 2 my led to the isolation of populations in the upper Y.R. and L.T.R. drainages (see *Hewitt, 1996*). During glacial periods and low sea levels, suitable habitat for Yazoo Darters would have expanded downstream, perhaps into the present Mississippi Alluvial Plain when sea levels were at their lowest (90–140 m below present). Streams were smaller (less precipitation), may have been entrenched in bedrock, and had higher gradients, coarse substrate, and cold, clear water. During interglacial periods sea levels rose, streams had more water, gradients moderated, stream valleys filled in with fines, and streams were no longer confined to bedrock. Suitable habitat for Yazoo Darters would have moved upstream.

As suitable habitat shifted up- and downstream in the Yazoo River Basin, connectivity among groups of Yazoo Darters in tributary streams would also have changed. During interglacial periods when streams were not confined to bedrock, changes in stream configurations seem more likely, especially in headwaters. However, during glacial periods, streams were smaller (climate was much drier) and confluences lower in the watershed were less likely to be barriers to dispersal because they were smaller and may have been suitable habitat for Yazoo Darters. Dispersal among tributaries under these conditions seems more likely (*Fisk, 1944*; *Rittenour, Blum & Goble, 2007*; *Past Interglacials Working Group of PAGES, 2016*).

Spatial changes in the downstream extent of suitable habitat likely interacted with changes in the location of the ancient confluence of the Y.R. and L.T.R. to isolate Yazoo Darter populations. Reliable data exists for estimating the number, duration, and timing of glacial and interglacial periods over about the last 800,000 years (*Past Interglacials Working Group of PAGES, 2016*). An estimated 11 cycles between glacial and interglacial periods are identified. Estimated duration of interglacial periods is much shorter (166,700 years) than glacial and transitional periods (633,300 years) (*Fisk, 1944*; *Past Interglacials Working Group of PAGES, 2016*). Given this setting, downstream connectivity among demes would have likely had greater influence structuring Yazoo Darter populations between the two major drainages than possible shifts in stream configurations. Further, changes to the position of the confluence of the L.T.R. and Y.R. and with the ancient predecessors of the Ohio and Yazoo rivers were likely instrumental in the phylogenetic pattern seen in our results (see text and figures in (*Fisk, 1944*); Fig. 1). It seems apparent that at some point (about 0.4–0.8 my) suitable habitat for Yazoo Darters no longer encompassed the confluence of the Y.R. and L.T.R. during glacial periods.

Our results help to refine the management actions (i.e., human-assisted gene flow) suggested in *Sterling et al. (2012)*. Phylogenetic trees show a weak to moderate association between watersheds and clades within drainages, though we did not find clear evidence of genetically distinct groups that were consistent with geography (e.g., management units to guide human-assisted gene flow) (Figs. S4–S7). Even so, based on our results and those in *Sterling et al. (2012)*, we recommend relocation of individuals among tributaries that are closest together within watersheds as categorized in *Sterling et al. (2012)*. Within the Y.R. watershed we recommend restricting movement of individuals to within the Otoucalofa Creek watershed or adjacent tributaries to the mainstem Y.R. Since genetic diversity was higher in the L.T.R. than in the Y.R., and because populations in the Y.R. face greater

risks (i.e., area of distribution is much smaller, estimates of effective population sizes are extremely low, there is no evidence of contemporary gene flow among adjacent tributaries, streams yielding Yazoo Darters are nearly all on private lands, and there is rapid urban development in this drainage, see *Sterling et al., 2012*; *Sterling, Warren Jr & Henderson, 2013*), human-assisted gene flow within the Yocona River should be implemented. Research aimed at identifying mechanisms of gene flow is also desperately needed for the Yazoo Darter, which would also help inform management of other imperiled forms of snubnose darters.

## CONCLUSIONS

Our results indicate that populations of the Yazoo Darter in the Y.R. drainage are genetically distinct and represent a recently diverged and undescribed cryptic species of snubnose darter. However, because phylogenetic evidence constitutes only one line of evidence for divergence, we recommend that other lines of evidence for species delimitation under the unified species concept (*de Queiroz, 2007*) be examined. Though there are no obvious differences in pigment patterns or color between the populations in each drainage, *Suttkus, Bailey & Bart Jr (1994)* noted modal differences in lateral line scale counts and *Sterling, Warren Jr & Henderson (2013)* showed that Yazoo Darters in the Y.R. drainage are significantly longer than those in the L.T.R. drainage. Further investigation of morphology, meristics, and pigment patterns is warranted.

## ACKNOWLEDGEMENTS

We would like to thank M Bland, Z Barnett, A Carson, C Smith, B Sterling, W Sterling, and G McWhirter for help collecting samples. We are also grateful to J Hubbell and J Schaefer for sharing data and G Henderson for help with figures. Any use of trade, firm, or product names is for descriptive purposes only and does not imply endorsement by the US Government.

### Funding
Funding was provided by the USDA Forest Service, Southern Research Station, Center for Bottomland Hardwoods Research. The funders had no role in study design, data collection and analysis, decision to publish, or preparation of the manuscript.

### Grant Disclosures
The following grant information was disclosed by the authors:
USDA Forest Service, Southern Research Station.
Center for Bottomland Hardwoods Research.

### Competing Interests
The authors declare there are no competing interests.

## Author Contributions

- Ken A. Sterling and Stuart V. Nielsen conceived and designed the experiments, performed the experiments, analyzed the data, prepared figures and/or tables, authored or reviewed drafts of the paper, and approved the final draft.
- Andrew J. Brown conceived and designed the experiments, performed the experiments, analyzed the data, authored or reviewed drafts of the paper, and approved the final draft.
- Melvin L. Warren, Jr. conceived and designed the experiments, authored or reviewed drafts of the paper, and approved the final draft.
- Brice P. Noonan conceived and designed the experiments, performed the experiments, authored or reviewed drafts of the paper, and approved the final draft.

## Animal Ethics

The following information was supplied relating to ethical approvals (i.e., approving body and any reference numbers):

University of Mississippi IACUC Committee approved the study (IACUC protocol 09-027).

## Field Study Permissions

The following information was supplied relating to field study approvals (i.e., approving body and any reference numbers):

Stream sampling and fish collections were approved by the Mississippi Museum of Natural Science under a general permit for all our scientists and technicians for all field studies. Annual fish collection permits from 2009-2017: 0604091, 0513101, 0624112, 0622122, 0602132, 0610142, 0624151, 0715163, 1010173.

## Data Availability

The genetic sequences are available at GenBank (see Tables. S1–S4).

## Supplemental Information

Supplemental information for this article can be found online at http://dx.doi.org/10.7717/peerj.9014#supplemental-information.

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
