# Peer review of "Cryptic diversity among Yazoo Darters (Percidae: Etheostoma raneyi) in disjunct watersheds of northern Mississippi"

_PeerJ, doi:10.7717/peerj.9014_

## Round 0.1 · original submission · Major Revisions

This is a solid, relatively straightforward paper that is concise and to the point. Although the genetic data set is not extensive, it does include both mitochondrial and nuclear loci. All three reviewers were generally enthusiastic about the manuscript but pointed out a number of issues that need to be addressed before it is suitable for publication. I would like to ask that you carefully consider all of the written comments and those in the attached annotated manuscripts. Your cover letter should thoroughly document how each criticism was addressed or provide a substantive argument for not doing so. A few comments from me below.

1) I like the explicit use of species concept but am a bit puzzled as to why you didn’t go ahead and formally recognize the new species. I would strongly encourage you to do so.
2) I agree with the reviews that some of the phylogenetic trees need to be pulled from the supplemental material to the main manuscript. I would also encourage you to consider publishing an actual image of the fish in the main manuscript (maybe combined with the supplemental habitat figures).
3) Please include the likelihood values for your trees in the figure legend – this ensures repeatability.
4) As noted by one of the reviewers some of the support values are actually very low and shouldn’t be characterized as “moderate.”

Reviewer 1 ·

Basic reporting

The Yazoo Darter, Etheostoma raneyi, has an incredibly small distribution in Mississippi waters. Microendemism is common among darters, and limited distributions are at risk of extirpation due to anthropogenic influences. The topic of the manuscript is relevant to conservation genetics as well as reintroduction or restoration efforts. Previous work on this group included the use of mitochondrial markers (Powers and Warren, 2009) where the authors recovered two clusters of Yazoo Darters and microsatellite data (Sterling 2012) that showed genetically distinct populations in each stream. Based on the limited range of the species, its imperiled status, and evidence of distinct genetic lineages, the authors focus on the Yazoo River system and increase taxon sampling to investigate cryptic diversity in the drainages of the upper Yazoo River.

The methods implemented in this paper are appropriate for assessing the genetic variability within the Yazoo Darter range; however, the presentation and assessment of these methods muddles the question at hand. The introduction states the goal is to understand and elaborate on putative cryptic diversity between the Yocona and the Little Tallahatchie Rivers previously reported by Powers and Warren; however, the discussion focuses on reviewing the diversity within rivers as reported by Sterling 2012. Congruence between the introduction/stated goals and the conclusions drawn from analyses is important for conveying the primary message of this paper. The authors should reflect on the primary goals of the paper and ensure the main message is being appropriately communicated and supported with methods and discussion.

The figures of this paper are not entirely relevant to the content of the article. Figure captions should be expanded to appropriately describe their content.
Figure 1 shows the distribution of Yazoo darters as well as two other Adonia clade species and the “Spring Creek” Darter. After reading the manuscript, I do not understand what this figure contributes towards the paper. There is little to no discussion of E. pyrrhogaster, E. cervus, or “Spring Creek” Daters, and what discussion is present is not relevant to the question of genetic diversity within the Yazoo River. The authors may choose to expand their discussion of these other taxa or remove the first figure from the paper.

Figure 2 should be the primary map for the manuscript. Portions of Figure 1 such as the line of the Upper Gulf Coastal Plain should be included in Figure 2. The caption associated with this figure (and the other figures) needs to be expanded to cover everything presented within it. A good portion of the Materials and Methods (lines 99-106) is dedicated to explaining the sampling area, yet none of the rivers or streams are labeled on the map. In GIS, you can highlight/shade the drainages by HUC (hydrologic unit code). I would recommend coloring/shading the watersheds consistently with the colors presented in Figure 4 and making the numbered boxes and circles white. Alternatively, the authors could color the numbered boxes and circles consistent with Figure 4.

Considering the amount of heterogeneity between the two analyses, the authors should concatenate their cytb and S7 data. I do not like that the S7 tree, which tells a different story than the cytb tree, is delegated to the supplemental material rather than presented and discussed in the main paper.

Figure 4 is fine with the nodes collapsed, given that the authors discuss the lack of internal support for the nodes; however, it may be useful to have the full trees (cytb, s7, and a concatenated tree) included in the supplementary materials

The authors spend some time discussing cryptic diversity and molecular methods used to understand population- and species-level variation. A few recent papers are provided that the authors may find relevant to their discussion:
Geheber 2019: Contemporary and Historical Species Relationships Reveal Assembly Mechanism Intricacies among Co-occurring Darters (Percidae: Etheostomatinae)
Hayes and Piller 2019: Patterns of diversification in a North American endemic fish, the Blackbanded Darter (Perciformes, Percidae)
Near et al., 2016: Systematics and taxonomy of the snubnose darter, Etheostoma simoterum (Cope)
Wagner and Blanton 2017: Do River Drainage Boundaries Coincide with Phylogeographic Breaks in the Redband Darter?

The call for human-assisted gene flow presented in Sterling 2012 is further supported by the genetic analysis presented here. The results of this paper are important for conservation, stream remediation, and population supplementation. With edits to the manuscript listed above, this paper will be a good contribution to the body of knowledge associated with the Yazoo Darter.

Experimental design

The methods implemented in this paper are appropriate for assessing the genetic variability within the Yazoo Darter range; however, the presentation and assessment of these methods muddles the question at hand.

Two portions of the methods require further explanation and elaboration. First, the molecular clock analysis is not sufficiently described (lines 150-152) for reanalysis by future researchers and the results are not presented in the results section. Considering a good portion of the discussion revolves around the timing of glacial cycles and divergence time estimates of the two streams, this needs to be strengthened. The second is the statement in lines 194-195 regarding the use of pairwise genetic sequence divergence. This reason for comparing across all Adonia clade species needs to be needs to be elaborated upon and the statement at line 194 particularly should be justified.

Validity of the findings

The call for human-assisted gene flow presented in Sterling 2012 is further supported by the genetic analysis presented here. The results of this paper are important for conservation, stream remediation, and population supplementation. With edits to the manuscript listed above, this paper will be a good contribution to the body of knowledge associated with the Yazoo Darter.

Additional comments

Other comments are provided in the attached .pdf file.

Annotated reviews are not available for download in order to protect the identity of reviewers who chose to remain anonymous.

Reviewer 2 ·

Basic reporting

1. Clear and unambiguous English
The writing is overall clear and the English is strong.

2. Sufficient Literature/Background
Background in the introduction is thorough, targeted to the Yazoo region and the biology of the darter species being examined.

3. Professional Article Structure
The article contains all the elements of a scientific paper.

4. Results relevant to hypothesis
While the authors don’t have a discrete hypothesis (they are instead drawing inferences form observed genetic diversity), the results presented are relevant to their research goals.

Experimental design

1. Original primary research
The research goal, date collection, methods of analysis, and presentation or results are all consistent with a primary research article.

2. Well defined research question
The research goal was to investigate cryptic diversity within the Yazoo darter using mitochondrial and nuclear gene sequence data.

3. Rigorous investigation
This is a very targeted study, with limited scope and conclusions. That being said, while none of the methods or analyses here are cutting edge, they are consistent with current practice in investigations of intraspecific genetic variation in fish species with limited range.

4. Sufficient methods
The methods utilized are appropriate for the question

Validity of the findings

1. Benefit to literature/impact
While it is limited in scope, the paper serves as a valuable baseline analysis of the population genetic structure of this darter.

2. All underlying data have been provided and are robust
I detected no issues.

3. Conclusions are clear, relate to hypothesis, and limited to results
Overall yes, but see my comments below.

4. No unwarranted speculation
There is some speculation in the discussion, but this is clearly presented as a ‘proposition.’

Additional comments

It would be helpful to indicate the watersheds sampled within each drainage on lines 102-106 somewhere on the map on Fig.1, or probably better on Fig. 2.

Line 160, the authors that reciprocal monophyly was ‘moderately’ supported by the Bayesian analysis. In order for support to be ‘strong’, both nodes would need to be supported by 0.95 or higher (a somewhat arbitrary benchmark). However, since one of the has a posterior probability of 0.12 (indicating only 12% of the sample trees recovered that clade), the term ‘moderate’ is a bit misleading. ‘Moderate’ might mean one of the nodes supported by 0.9, for example. These data suggest no or little support for reciprocal monophyly in the Bayesian analysis given that support for both nodes is required.

Lines 161-162. “Phylogenetic trees generated using the S7 gene generally lacked support for samples from the Little Tallahatchie…” It is not clear what object lacked support. The sentence makes is sound as though samples lacked support. A sample can’t really be supported (or not) by the phylogeny. Is the statement meant to imply that the phylogeny based on the S7 data lacked support for a monophyletic clade corresponding with Little Tallahatchie samples?

It is not clear why the trees based on the S7 sequence data were not shown in a figure in the main body of the manuscript. These tree could be included in Figure 3 with the mitochondrial tree (parts A, B, and C for example).

Figure 3 – Are the bubble sizes for LT and YR proportional to the number of individuals sampled or the total number of unique haplotypes?

The authors mention support for the recognition of two species based on the phylogenetic species concept (Lines 181-183), however no specific data are referenced in support of this assertion. Nixon and Wheeler (1990) define a phylogenetic species as "the smallest aggregation of populations (sexual) or lineages (asexual) diagnosable by a unique combination of character states in comparable individuals (semaphoronts).” Are the authors suggesting that diagnosable states are listed in table 2? While these represent discrete characters, the reconstructed phylogenies do not provide strong, consistent support for two distinct clades. Support varies by locus and method of analysis (Bayesian vs. Likelihood). While there is evidence of divergence, raising the potential for two different species is a significant claim and should be clearly supported by evidence. However, very little explanation in this subject is provided in the discussion despite it being a lead point in the section. Given the current understanding of the distinction between species trees and gene trees, the authors might consider modifying the language to suggest the potential for species recognition, barring further analysis of data, both genetic and morphological. If not, there needs to be a much deeper explanation of how the data support the assertion and a nuanced discussion of these data considering known differences between gene tree and species tree inference.

Overall, the authors have drawn some otherwise sound conclusions about the genetic distinctiveness of the populations in these two drainages and made a good case for within drainage translocations. Conclusions regarding species recognition extend a bit beyond the data. Studies such as this focusing on population genetics of vulnerable microendemics are critical to furthering our understanding of how to manage them. This is a strong paper and should be published.

·

Basic reporting

The manuscript meets basic reporting requirements; in several cases statements require clarification and figures require revised labels or heading information, but these have been noted with specific comments in the text.

Experimental design

The manuscript meets the the expectations of experimental design and fits the aims and scope of the journal. In a few cases, additional information is required to clarify minor points regarding methodology and these are noted with specific comments in the text.

Validity of the findings

Findings and conclusions of the manuscript are supported by the data/results presented.

Additional comments

The authors use DNA sequencing of populations of the Yazoo Darter, E. raneyi, to examine phylogeographic relationships and to identify cryptic diversity in this species. The writing of the manuscript is overall clear and effective, the methodology sufficient, and the conclusion are supported by the data presented. They find support for cryptic diversity in that populations from the Yocona River represent a distinct clade and taxon of E. raneyi, which they refer to as the "Yocona Darter". This work makes a good contribution to our understanding of diversity of southeastern US fishes and makes conservation recommendations based on their findings for the focal taxa, which are much needed.

One of the primary concerns that should be addressed by the authors is related to how undescribed species are recognized in the manuscript. Specific comments related to this are provided throughout the text. One, use of names like "Spring Creek" Darter (as in Figure1) is problematic because many readers would have no idea what taxon this refers to; in later tables this is identified as a member of the E. zonistium or Bandfin Darter species group. I recommend labeling as E. cf. zonistium - Spring Cr (or Creek) instead for clarity. I also think using "Yocona Darter" in the text, tables and figures is problematic as this implies an official common name for the taxon which is proposed as a new species, but has not been officially described. This can create taxonomic confusion and could be confusing to the readers that might skim the article, or look at figures to get a quick take-away. I recommend referring to as E. cf. raneyi - Yocona R. (or Yocona River form of E. raneyi). The data clearly support that it is distinct and its not an issue to note and identify that the data support it as a distinct species, however. Also see the use of such annotation for members of the E. bellator and E. brevirostrum species groups, where "common names" are used for undescribed species without . Related to names and population identifiers, there is some inconsistency in use across text, tables and figures that should be addressed as well. I also recommend using E. raneyi rather than "Yazoo Darter" in the text, tables, figures, throughout (or rather than switching back and forth).

I like that the authors use this data to offer more informed conservation actions, however, I think that paragraph could be refined with more specific and clear recommendations based on the totality of all the data that is now available for the Yazoo Darter. Specific comments are in the text.

In general, I think the Figure legends would benefit from a more detailed description of what is shown and consistent labeling of taxa and watershed-drainages or "clades" could be improved and make it easier for the reader to follow the big picture.

There are other minor issues, questions, and points of clarification that should be reviewed and addressed as noted with comments in the attached pdf.

---

## Round 0.2 · accepted · Accept

The authors have done a superb job addressing each of the reviewers concerns. It was already a very solid paper but, as pointed out by the authors, nicely improved via really thoughtful feedback from colleagues. I am not entirely sure what the reviewer's concern was with the informal clade name 'Adonia' - it is entirely up to your discretion if you'd like to retain that name - my understanding is that it's an informal clade name and doesn't really present any nomenclatural problems.